# AgRP1 modulates breeding season-dependent feeding behavior in female medaka

Yurika Tagui[1], Shingo Takeda[1], Hiroyo Waida[1], Shoichi Kitahara[1], Tomoki Kimura[2], Shinji Kanda[3], Yoshitaka Oka[1], Yu Hayashi[1], Chie Umatani[1,4]*

[1]Graduate School of Science, The University of Tokyo, Tokyo, Japan; [2]Department of Applied Physics, The University of Tokyo, Tokyo, Japan; [3]Atmosphere and Ocean Research Institute, The University of Tokyo, Chiba, Japan; [4]Graduate School of Agriculture, Tokyo University of Agriculture and Technology, Tokyo, Japan

*For correspondence:
chie@go.tuat.ac.jp

## eLife Assessment

This article provides **fundamental** new insight into the mechanisms linking photoperiod, reproduction function, and feeding activity, using medaka, a genetic model that itself exhibits photoperiodic responses. As well as identifying key neuropeptide genes that are regulated by photoperiod and involved in regulating feeding activity, the authors establish a knockout line for agrp1 using CRISPR Cas9-based approach, profiting from the extensive use and development on this methodology in medaka. The combination of the RNAseq and quantitative in situ hybridization analysis with the knockout results as well as the study of ovariectomized fish provides **compelling** evidence implicating agrp1 in feeding regulation in response to photoperiod and reproductive status.

**Abstract** Feeding and reproduction are known to be closely correlated with each other, and the seasonal breeders show breeding season-dependent feeding behavior. However, most model animals do not have definite breeding seasonality, and the mechanisms for such feeding behavior remain unclear. Here, we focused on female medaka (*Oryzias latipes*); they show breeding season-dependent feeding behavior, and their condition of breeding season can be experimentally controlled by day-length. We first demonstrated that, among previously reported feeding-related peptides (neuropeptides involved in feeding), agouti-related peptide 1 (*agrp1*) and neuropeptide y b (*npyb*) show higher brain expression under the breeding condition than under the non-breeding one. Combined with analysis of *agrp1* knockout medaka, we obtained results to suggest that long day-induced sexually mature condition, especially ovarian estrogenic signals, increase the expressions of *agrp1* in the brain, which results in increased food intake to promote reproduction. Our findings advance the understanding of neural mechanisms of feeding behavior for reproductive success.

## Introduction

Feeding behavior is essential to animals for their survival and reproduction and is known to be modulated by various internal and external factors: nutritional status, sexual maturity, temperature, seasonality, etc. This behavior is known to be closely correlated with reproduction (*Kauffman and Rissman, 2004*), which is an essential biological activity important for the animal life. Previous studies reported that nutritional-state modulates reproductive behaviors and functions (*Chen et al., 2006*; *Amirjani et al., 2019*; *Volk et al., 2017*; *Lynn et al., 2010*). For example, musk shrews show defective sexual behavior under fasted conditions (*Temple and Rissman, 2000*). In addition, not a few

studies demonstrated that fasting-induced low energy condition suppresses reproduction (*Evans and Anderson, 2012*; *Kalra and Kalra, 1996*; *Kirkwood et al., 1987*; *Merry and Holehan, 1979*; *Hasebe et al., 2016*). Thus, it has been well investigated how nutritional status resulting from feeding modulates reproduction. On the other hand, it has been reported that some animals change their feeding behavior during the breeding season. For instance, the black seabream migrates to the shallow water during the breeding season (*Tsuyuki, 2018*; *Kawai et al., 2020*) where they can get more food, and the white-tailed deer spends more time for feeding under reproductive status (*Stone et al., 2017*). Such a close relationship between reproduction and feeding is thought to be important for biological fitness. However, the regulatory mechanisms for breeding season-dependent feeding behavior are still unknown. One possible reason is that most of the model animals appear to have lost the well-defined breeding season. Although the mammalian models, mice and rats, and teleost model zebrafish, have reproductive cycles of about 4–5 days (*Nilsson et al., 2015*; *Peute et al., 1978*), they do not have definite breeding seasonality. Thus, the mechanisms for long-term changes in feeding behavior according to the breeding season have not yet been studied in detail.

Here, as a model animal for the seasonal breeder, we used a teleost fish, medaka (*Oryzias latipes*). Medaka is a useful model animal, whose reproductive status can be experimentally controlled by day-length (*Robinson and Rugh, 1943*; *Egami, 1954*) while keeping an appropriate temperature constant. In the long-day (LD) condition (14 h light/10 h dark), female medaka becomes reproductive and regularly spawns every day, while it becomes non-reproductive in the short-day (SD) condition (10 h light/14 h dark). In other words, LD or SD condition can induce breeding or non-breeding season of female medaka, respectively. Thus, medaka enables us to analyze the mechanisms of breeding season-dependent feeding behavior without consideration for possible changes in metabolism and gene expressions due to the changes in ambient temperature, which means medaka is suitable for this study.

Regulatory mechanisms of feeding behavior have mainly been analyzed in mammals. These studies reported that some neuropeptides, such as agouti-related peptide (AgRP) and neuropeptide Y (NPY), are involved in the control of feeding and called 'feeding-related peptides' as key molecules for the regulation of feeding behavior (*Hahn et al., 1998*; *Aponte et al., 2011*; *Krashes et al., 2011*; *Andermann and Lowell, 2017*). Teleosts have also been thought to possess a regulatory mechanism for feeding similar to mammals. In fact, expression of homologous genes coding for feeding-related peptides have been reported (*Rønnestad et al., 2017*; *Conde-Sieira and Soengas, 2016*). On the other hand, although administration of some of them have been suggested to induce feeding behavior in teleosts as well (*Rønnestad et al., 2017*), their functions in feeding behaviors still remain unclear.

In the present study, to understand mechanisms of breeding season-dependent feeding behavior, we focused on female medaka, which clearly show seasonal changes in breeding conditions by day length (*Mitani et al., 2010*; *Kanda et al., 2008*) under the fixed appropriate temperature. We first quantified changes in feeding behavior according to the breeding states and found that female medaka under the condition of breeding season (LD) eat more than those under the condition of non-breeding (SD). Therefore, we searched for genes that show breeding state-dependent changes in expression and found some candidates for feeding-related peptides in medaka. We then analyzed expressions of the candidate genes by using RNA-seq, in situ hybridization (ISH), and RT-qPCR, and analyzed phenotypes of gene knockout medaka. These results led us to conclude that AgRP1 plays a key role in the breeding season-dependent changes in feeding behavior via ovarian estrogenic signals.

## Results

### Feeding behavior of female medaka is upregulated in the breeding season

To analyze food intake of male and female medaka in breeding/non-breeding seasons, we first established a method for measuring food intake in medaka. In brief, we placed medaka in a white cup, fed brine shrimp to medaka in all-you-can-eat style for 10 min, and counted the leftover brine shrimp in the cup with a 'shrimp-counter' system (called Japanese 'Wanko-soba'-like method, *Figure 1—source code 1* and *Figure 1—figure supplement 1*). We used this system to analyze food intake of male and female medaka under the breeding condition equivalent to that in the breeding season (kept under LD condition) or under the non-breeding condition equivalent to that in non-breeding season

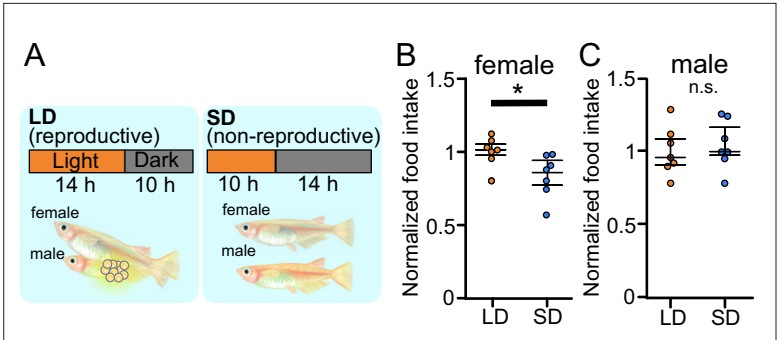

**Figure 1.** Reproductive female medaka show larger amount of food intake. (**A**) Light conditions for breeding and non-breeding status. (**B**) Food intake (10 min) of female medaka in long day (LD) (orange; n=7) and SD (blue; n=7) conditions, normalized to the amount of artemia eaten by medaka in LD (breeding) condition (p=0.02519, *U*=42.5). (**C**) Food intake (10 min) of male medaka in LD (orange; n=7) and short day (SD) (blue; n=7) conditions, normalized to the amount of artemia eaten by medaka in LD (breeding) condition (p=0.6540, *U*=20.5). Mann–Whitney *U* test, *p<0.05. n.s., not significant.

The online version of this article includes the following source data, source code, and figure supplement(s) for figure 1:

**Source code 1.** The code of 'Shrimp-counter' system.

**Source data 1.** The numerical data for *Figure 1*.

**Figure supplement 1.** 'Wanko-soba' method for calculation of the amount of food intake of fish.

**Figure supplement 2.** Whole-brain gene expression in long day (LD) (n=3) and short day (SD) (n=3) female medaka.

**Figure supplement 2—source data 1.** The numerical data for *Figure 1—figure supplement 2*.

(SD condition) (*Figure 1A*; *Kanda et al., 2008*). We found that female medaka under the breeding (LD) condition eat more than those under the non-breeding (SD) condition (*Figure 1B*; p=0.02519). In contrast to female, in males there was no significant difference in food intake between the breeding and non-breeding condition (*Figure 1C*; p=0.6540). Since these results demonstrated that females, not males, show breeding season-dependent feeding behavior, we focused only on female medaka in the following analyses on neuronal mechanism for breeding season-dependent feeding behavior. Next, to examine which gene products modulate feeding behavior of female medaka in the breeding season, we performed mRNA-sequencing (RNA-seq) using the whole brain of female medaka in breeding condition (LD) and non-breeding condition (SD) (*Figure 1—figure supplement 2A*). Overall, 1025 genes showed significantly different expression between LD and SD female medaka. *Figure 1—figure supplement 2B* shows a heat map of representative genes mainly related to neuroendocrine system, which were differently expressed between LD and SD females. Among the conventional candidate feeding-related neuropeptides, we identified two kinds of neuropeptides, *agrp1* and *npyb*, both of which showed higher expression in LD than in SD (*Figure 1—figure supplement 2C and D*). Both AgRP and NPY are known to have orexigenic effects mainly in mice (*Schwartz et al., 2000*; *Andermann and Lowell, 2017*). Therefore, in the subsequent analyses, we focused on *agrp1* and *npyb* as candidate genes that modulate breeding season-dependent feeding behavior in female medaka.

## AgRP1, NPYa, and NPYb may be the 'feeding-related peptides' in female medaka

Medaka has two *agrp* paralogues, *agrp1* and *agrp2,* and two *npy* paralogues, *npya* and *npyb*, which arose from third round whole genome duplication early in the teleost lineage (*Liu et al., 2019*; *Sundström et al., 2008*). Therefore, we next examined the anatomical distribution of neurons expressing *agrp1*, *npyb*, and their paralogs in the female brain by in situ hybridization (ISH). We found that *agrp1*- and *npyb*-expressing neurons are distributed in local brain regions (*Figure 2A, B and E*), while *npya*- and *agrp2*- expressing neurons are more widely distributed (*Figure 2A, C and D*) in the brain. The *agrp1* neurons were distributed in the nucleus ventralis tuberis (NVT) of the hypothalamus (*Figure 2B*), while *agrp2* neurons were expressed in the telencephalon and in the hypothalamus

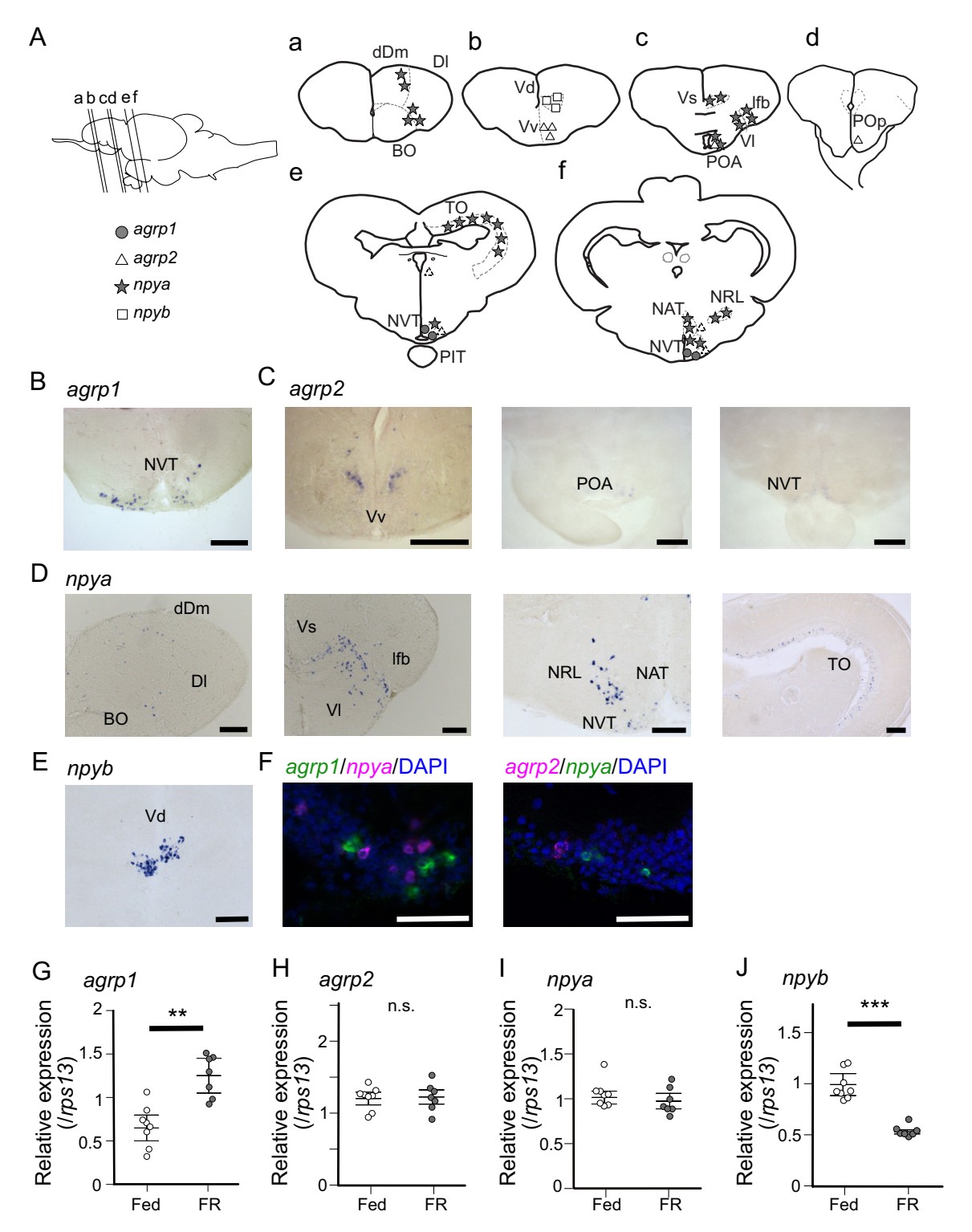

**Figure 2.** *agrp1*- and *npyb*-expressing neurons are distributed locally, while npya- and *agrp2*-expressing neurons are distributed more widely in the brain. (**A**) Illustration of the lateral view of medaka brain and distributions of cell bodies expressing each *agrp* or *npy* gene. The oblique lines labeled with (**a–e**) indicate the level of the frontal sections in (**a–e**). (**a–e**) Illustrations of frontal sections showing the distribution of neurons expressing each gene. The localization of neurons is indicated in the right half of the illustrations. BO, bulbus olfactorius; dDm, dorsal region of area dorsalis telencephali pars medialis; Dl, area dorsalis telencephali pars lateralis; lfb, lateral forebrain bundle; NAT, nucleus anterior tuberis; NRL, nucleus recessus lateralis; NVT, nucleus ventralis tuberis; PIT, pituitary; POA, area preoptica; POp, nucleus preopticus pars paravocellularis; TO, tectum opticum; Vd, area ventralis

*Figure 2 continued on next page*

*Figure 2 continued*

telencephali pars dorsalis; Vl, area ventralis telencephali pars lateralis; Vs, area ventralis telencephali pars supracommissuralis; Vv, area ventralis telencephali pars ventralis; *agrp1* (gray circle), *agrp2* (open triangle: high expression, dotted triangle: low expression), *npya* (star), *npyb* (open square). (**B**) *agrp1*-expressing neurons are localized in NVT. Scale bar: 100 µm. (**C**) *agrp2*-expressing neurons are observed in Vv, POA, and NVT. Scale bar: 100 µm. (**D**) *npya*-expressing neurons are distributed in dDm, Dl, BO, Vs, Vl, lfb, NRL, NAT, NVT, and TO. Scale bar: 100 µm. (**E**) *npyb*-expressing neurons are localized in Vd. Scale bar: 100 µm. (**F**) Left: *agrp1* (green) and *npya* (magenta) are distributed in NVT, but the two genes are not co-expressed. Right: *agrp2* (magenta) and *npya* (green) are distributed in NVT, but the two genes are not co-expressed. Scale bars: 50 µm. (**G–J**) *agrp* and *npy* expressions in the whole brain of female medaka under normally fed condition (Fed; white; n=7) or 2-week food restricted (FR; gray; n=8). (**G**) *agrp1* (p=0.001243, U=2), (**H**) *agrp2* (p=0.9551, U=29), (**I**) *npya* (p=0.2319, U=39), and (**J**) *npyb* (p=0.0003108, U=56). Mann–Whitney *U* test, **p<0.01, ***p<0.001. n.s., not significant.

The online version of this article includes the following source data and figure supplement(s) for figure 2:

**Source data 1.** The numerical data for *Figure 2*.

**Figure supplement 1.** The number of *npya*-expressing neurons and the expression of *agrp2* in the hypothalamus of fed and food-restricted (FR) medaka.

**Figure supplement 1—source data 1.** The numerical data for *Figure 2—figure supplement 1*.

---

(*Figure 2C*). On the other hand, *npya* neurons were distributed more widely from telencephalon to hypothalamus (*Figure 2D*). *npyb* neurons were distributed locally in the nucleus ventralis telencephali pars dorsalis (Vd) of the telencephalon (*Figure 2E*).

In mice, *agrp* is known to be only expressed in the hypothalamus and mostly co-expressed with *npy* (*Hahn et al., 1998*), and these AgRP/NPY neurons are known to regulate mammal feeding behavior (*Shutter et al., 1997*; *Broberger et al., 1998*; *Ollmann et al., 1997*; *Takahashi and Cone, 2005*). In medaka, on the other hand, *agrp1* signals were not observed in *npya* neurons (*Figure 2F*, left), although the both genes were expressed in the hypothalamus. In addition, *agrp2* signals were not observed in hypothalamic *npya* neurons (*Figure 2F*, right), either. These results suggest that AgRP and NPY are not co-expressed in medaka. Since AgRP and NPY of medaka showed different expressing patterns compared with other animals such as mice, we examined whether they act as modulators of feeding. We divided female medaka in LD condition into two groups; one group was kept under normally fed condition (Fed), and the other was kept under 2-week food restricted condition (FR). We then analyzed whole-brain expressions of these four genes. RT-qPCR analysis demonstrated that 2-week FR increased the expression of *agrp1* (*Figure 2G*; p=0.001243) but decreased that of *npyb* (*Figure 2J*; p=0.0003108), suggesting that the two peptides are involved in feeding in an opposite manner. On the other hand, *agrp2* did not significantly change their expressions between Fed and FR conditions (*Figure 2H*; p=0.9551). Although *npya* did not significantly change their expressions between Fed and FR conditions (*Figure 2I*; p=0.2319), it may be possible that *npya* expression changed in a specific brain region, since *npya* neurons are widely distributed in various brain regions as described above (*Figure 2D*). Since it is suggested that NPY released from hypothalamic *npy*-expressing neurons controls food intake in mice (*Kohno and Yada, 2012*), we also examined the *npya*-expression in medaka hypothalamus by ISH. We counted *npya*-expressing neurons in each hypothalamic region and compared them between Fed and FR female medaka (*Figure 2—figure supplement 1*). We found that the total number of *npya*-expressing neurons in the hypothalamus was significantly larger in Fed compared with FR (*Figure 2—figure supplement 1A*; p=0.02857). Here, significant increase in cell number was observed in nucleus recessus lateralis (NRL) (p=0.02857) and nucleus anterior tuberis (NAT) (p=0.02857), but not in NVT (p=0.1143) (*Figure 2—figure supplement 1B–D*). On the other hand, the expression of *agrp2* did not show remarkable difference in the hypothalamus under food restriction or not (*Figure 2—figure supplement 1E*). Thus, the results suggest that AgRP1, NPYa, and NPYb may be the 'feeding-related peptides' in female medaka.

## Both *agrp1* and *npyb* show higher expression levels in LD than in SD female medaka

To further examine the result of RNA-sequencing (*Figure 1—figure supplement 2*), we compared expression of *agrp1* and *npyb* between the female medaka under the breeding condition (LD) and those under the non-breeding condition (SD) using ISH and whole-brain RT-qPCR (*Figure 3*). First, we performed whole-brain RT-qPCR and found that the expression level of *agrp1* was higher in LD than in SD female (*Figure 3A*, p=0.001865). In ISH experiments, we observed larger number of

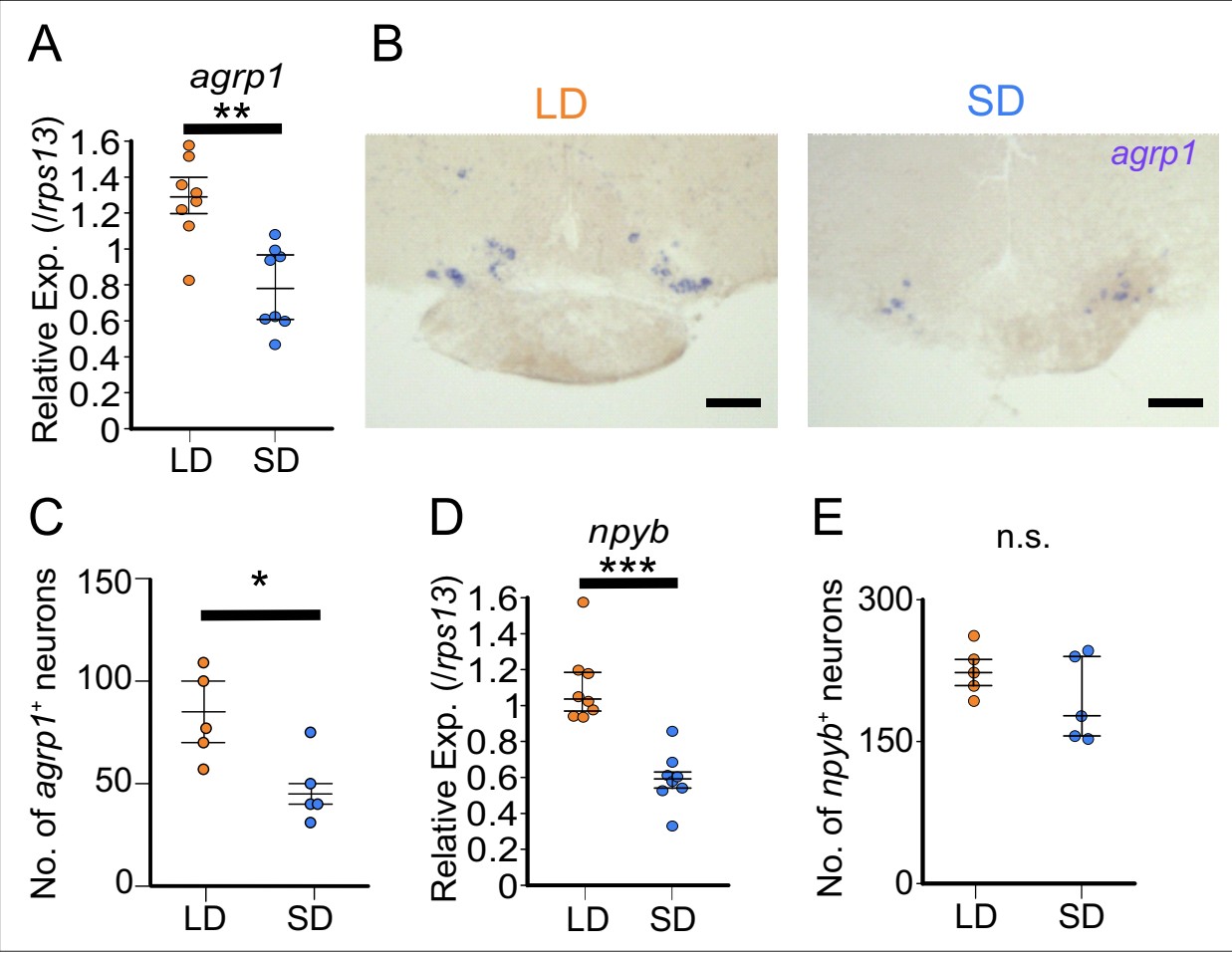

**Figure 3.** *agrp1* and *npyb* show higher expression levels in long day (LD) than in short day (SD) female. (**A**) *agrp1* expression in the whole brain of LD (orange; n=8) and SD (blue; n=8) female medaka (p=0.001865, *U*=60). (**B**) In situ hybridization (ISH) of *agrp1*-expressing neurons in LD and SD female medaka. Scale bar: 100 µm. (**C**) The number of neurons expressing *agrp1* in LD (orange; n=5) and SD (blue; n=5) (p=0.02828, *U*=23). (**D**) *npyb* expression in the whole brain of LD (orange; n=8) and SD (blue; n=8) female medaka (p=0.0001554, *U*=64). (**E**) The number of neurons expressing *npyb* in LD (orange; n=5) and SD (blue; n=5) (p=0.4206, *U*=17). The upper, middle, and lower bars show the third quartile, median, and the first quartile, respectively. Mann–Whitney *U* test, *p<0.05, **p<0.01, ***p<0.001. n.s., not significant.

The online version of this article includes the following source data and figure supplement(s) for figure 3:

**Source data 1.** The numerical data for *Figure 3*.

**Figure supplement 1.** Time course of the number of neurons showing in situ hybridization (ISH) signals for *agrp1*.

**Figure supplement 1—source data 1.** The numerical data for *Figure 3—figure supplement 1*.

*agrp1*-expressing neurons in LD than in SD females (*Figure 3B and C*; p=0.02828, *Figure 3—figure supplement 1*). Since the expression level of *agrp1* was higher in LD than that in SD (*Figure 3A*), higher expression of *agrp1* under the breeding condition may be due to the increase in the number of neurons expressing *agrp1*. On the other hand, *npyb* expression in RT-qPCR was significantly higher in LD than that in SD (*Figure 3D*; p=0.0001554), although ISH analysis demonstrated that *npyb*-expressing cell number was not significantly different between LD and SD (*Figure 3E*; p=0.4206). These results suggest that the expression level for each neuron increased in LD compared with SD. Thus, higher expression of *npyb* under the breeding condition may be due to the increase of expressions in each neuron expressing *npyb*.

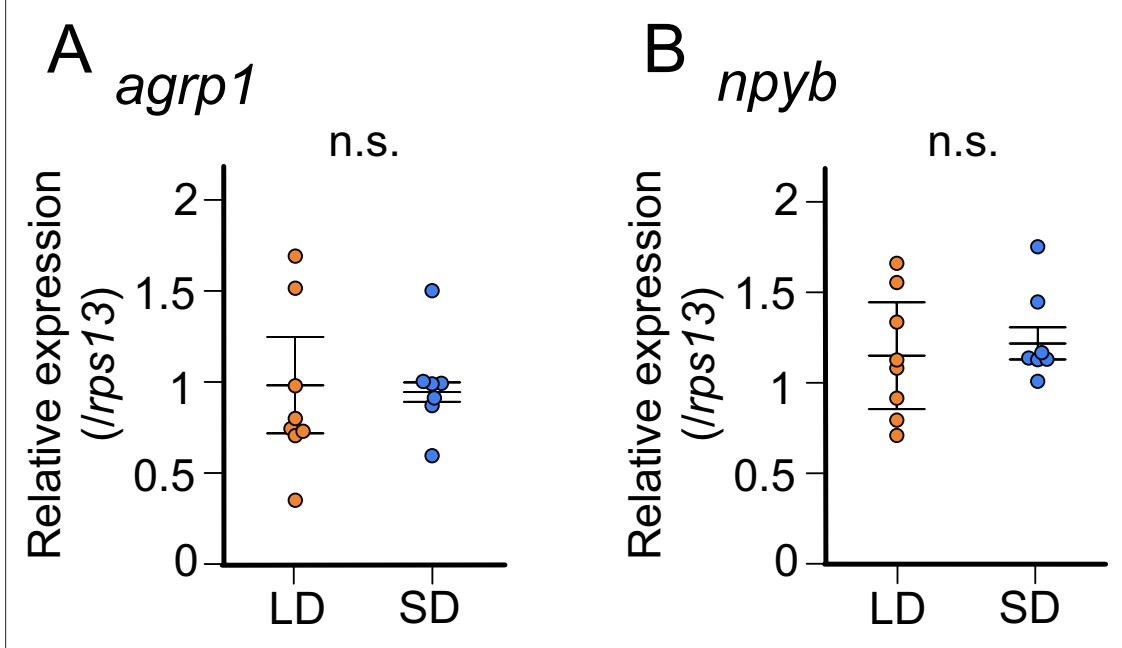

**Figure 4.** In juvenile female medaka, expression levels of neither *agrp1* nor *npyb* show significant change according to the day-length. (**A**) *agrp1* expression in the brain of juvenile female medaka (p=0.4179, *U*=21). (**B**) *npyb* expression in the brain of juvenile female medaka (p=0.3357, *U*=19). Long day (LD): orange, n=8; short day (SD): blue, n=7. The upper, middle, and lower bars show the third quartile, median, and the first quartile, respectively. Mann–Whitney *U* test. n.s., not significant.

The online version of this article includes the following source data and figure supplement(s) for figure 4:

**Source data 1.** The numerical data for *Figure 4*.

**Figure supplement 1.** Food intake of juvenile female does not show significant change according to the day-length.

**Figure supplement 1—source data 1.** The numerical data for *Figure 4—figure supplement 1*.

## In juvenile female medaka, expression levels of neither *agrp1* nor *npyb* show significant change according to the day-length

The results thus far indicates that expressions of *agrp1* and *npyb* are upregulated in female medaka under the condition of breeding season. Since the breeding/non-breeding state is experimentally controlled by day-length (LD/SD) in the present study, we examined which of the two factors, day-length itself or substance(s) from LD-induced mature ovary, regulates the expression of *agrp1* and *npyb*. Here, we used sexually immature juvenile medaka and compared their whole-brain expressions of *agrp1* and *npyb* under LD/SD conditions using RT-qPCR (*Figure 4*). We found that expression levels of neither *agrp1* nor *npyb* show significant difference between LD and SD (*Figure 4A* [p=0.4179] and *Figure 4B* [p=0.3357]). Furthermore, food intake of juvenile female was not different between LD and SD (*Figure 4—figure supplement 1*; p=0.7197). These results suggest that neither of them is regulated directly by day-length itself. Instead, the gene expression is suggested to be regulated by LD-induced sexual maturity.

## Estrogen, which is released from mature ovary, may affect the expression of *agrp1*

Among various factors associated with ovarian maturity, estrogens are known to be abundantly released from mature ovary and play important roles in reproductive readiness, sexual behavior, and so on *Jennings and de Lecea, 2020*; *Naftolin et al., 2007*; *Adachi et al., 2007*; *Clarkson and Herbison, 2009*; *Wintermantel et al., 2006*; *Micevych and Meisel, 2017*; *Melo and Ramsdell, 2001*. Among the ovarian estrogens, 17β-estradiol (E2) is the major factor important for reproduction (*Kanda et al., 2011*; *Kelly and Qiu, 2010*), and the blood E2 concentration of LD-conditioned female medaka is higher than those of SD (*Ikegami et al., 2022*). Thus, we hypothesized that E2 regulates the expression of *agrp1* and *npyb* under the condition of breeding season. We analyzed the expression

of *agrp1* and *npyb* in sham-operated (Sham), ovariectomized (OVX, fish with surgical ablation of the ovary), and OVX medaka with E2-administration (OVX+E) (*Figure 5A and B*). The OVX medaka were allowed to survive at least for 2 weeks to clear the endogenous E2 (*Kayo et al., 2020*), and spawning of all the Sham medaka were confirmed for three consecutive days. By using whole-brain RT-qPCR, we found that OVX induces significantly lowered *agrp1* expression than Sham, and OVX+E shows a tendency to recover *agrp1* expression compared with OVX (*Figure 5A*; Sham vs OVX: p=0.04310, OVX vs OVX+E: p=0.05790, Sham vs OVX+E: p=0.2000), which suggests that the ovarian E2 regulates *agrp1* expression. On the other hand, the expression levels of *npyb* did not show significant differences among the three groups (*Figure 5B*; Sham vs OVX: p=0.1386, OVX vs OVX+E: p=0.9991, Sham vs OVX+E: p=0.08120). In addition, food intake of OVX female was not significantly different between LD and SD (*Figure 5—figure supplement 1*; p=0.7308), which suggests that ovarian signal may be important for breeding season-dependent feeding behavior. Therefore, we focused more on the estrogenic regulation of *agrp1* expression.

Estrogens act mainly by interacting with estrogen receptors (*Chen et al., 2022*). Medaka has three kinds of estrogen receptors; *esr1*, *esr2a*, and *esr2b* (*Tohyama et al., 2016*; *Kinoshita et al., 2009*), and all of them have been reported to be expressed in NVT (*Zempo et al., 2013*), in which *agrp1* was also expressed (*Figure 2B*). Therefore, we examined co-expression of these *esr* genes and *agrp1*. As shown in *Figure 5C and D*, *esr2a* signal was clearly co-expressed in some *agrp1*-expressing neurons, which strongly suggests that E2 affects the expression of *agrp1* via *esr2a* in those neurons of NVT.

## *agrp1⁻ᐟ⁻* female medaka show a decrease in food intake and in the number of fertilized eggs

Our present experimental evidence thus far highlights the importance of *agrp1* as the factor modulating the season-dependent feeding behavior in medaka. To analyze the function of AgRP1 in medaka, we generated knockout medaka of *agrp1* (*agrp1⁻ᐟ⁻*) by using CRISPR/Cas9. The designed CRISPR guide RNA cleaved targeted sites of *agrp1* (exon3, *Figure 6—figure supplement 1A*), and we obtained *agrp1⁻ᐟ⁻* medaka, which has lots of amino acid changes in functional site for AgRP1 (*Figure 6—figure supplement 1B*). In *agrp1⁻ᐟ⁻* brain, AgRP1 immunoreactive signals, which were observed in WT, were not found (*Figure 6—figure supplement 1C*). These suggested that *agrp1⁻ᐟ⁻* possess nonfunctional AgRP. As for phenotype of the knockout, the *agrp1⁻ᐟ⁻* female medaka appeared skinny and the body weight was significantly lower than that of *agrp1⁺ᐟ⁺* (*Figure 6A*; body weight: p=0.04113; abdominal length: p=0.002165). In addition, abdominal height of *agrp1⁻ᐟ⁻* was also smaller than that of *agrp1⁺ᐟ⁺*, while the body length was not significantly different (*Figure 6A*; body length: p=0.3939). We next analyzed food intake of *agrp1⁻ᐟ⁻* female medaka in LD condition (breeding). As shown in *Figure 6B* (p=0.004329) and *Figure 6—figure supplement 2* (*agrp1⁺ᐟ⁺* vs *agrp1⁻ᐟ⁺*: p=0.5470, *agrp1⁺ᐟ⁺* vs *agrp1⁻ᐟ⁻*: p=0.009353, *agrp1⁻ᐟ⁺* vs *agrp1⁻ᐟ⁻*: p=0.01234), we found that LD *agrp1⁻ᐟ⁻* female medaka eat less than *agrp1⁺ᐟ⁺*. Then, we kept *agrp1⁻ᐟ⁻* medaka in LD or SD condition and compared their food intake. In contrast to *agrp1⁺ᐟ⁺*, *agrp1⁻ᐟ⁻* in LD condition did not show a significant increase in food intake compared with SD (*Figure 6C*; *p*=0.5953). We also examined whether loss of AgRP1 affects reproductive function. Whereas the *agrp1⁻ᐟ⁻* females were fertile, the pairs of *agrp1⁻ᐟ⁻* female and *agrp1⁺ᐟ⁺* male resulted in fewer spawned eggs than *agrp1⁺ᐟ⁺* pairs (*Figure 6D*; p=0.008658). In addition, the ovarian size of *agrp1⁻ᐟ⁻* appeared to be smaller than *agrp1⁺ᐟ⁺* (*Figure 6E*, left). In particular, since relative ovarian weight normalized by body weight (gonadosomatic index [GSI]) of *agrp1⁻ᐟ⁻* female tended to be marginally smaller than *agrp1⁺ᐟ⁺* (*Figure 6E*, right; p=0.06494), the smaller body size of *agrp1⁻ᐟ⁻* (*Figure 6A*) may drastically affect ovarian morphology. Since the number of spawned eggs was decreased in *agrp1⁻ᐟ⁻* female, we analyzed gene expressions of gonadotropins which should affect ovarian maturation. Oocyte maturation and ovulation are known to be regulated by gonadotropins, follicular-stimulating hormone (FSH) and luteinizing hormone (LH). As shown in *Figure 6F*, *agrp1⁻ᐟ⁻* females showed lower levels of expression of gonadotropin genes (*fshb* and *lhb; lhb:* p=0.008658; *fshb:* p=0.02597), which suggests that loss of function of *agrp1* impaired breeding season-dependent feeding behavior and led to attenuation of reproductive functions, especially the decrease in number of spawned eggs in the breeding season.

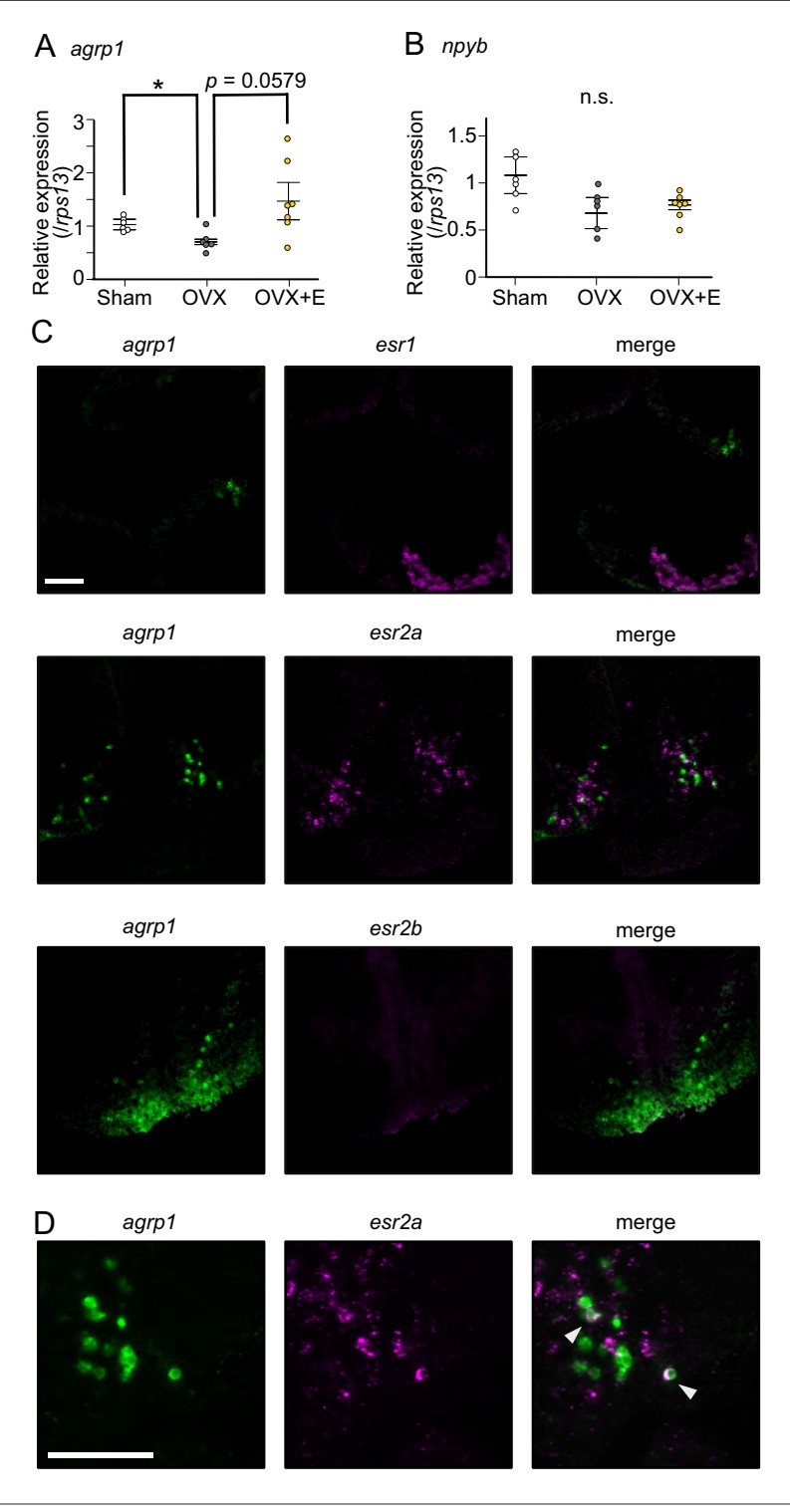

**Figure 5.** Estrogen, which is secreted from mature ovary, may affect the expression of *agrp1*. (**A**) Relative expression of *agrp1* in Sham (white; n=6), OVX (ovariectomized medaka, gray; n=6), and OVX+E (OVX medaka kept in the tank containing E2, yellow; n=7). Sham vs OVX: p=0.04310, OVX vs OVX+E: p=0.05790, Sham vs OVX+E: p=0.2000. (**B**) Relative expression of *npyb* in Sham, OVX, and OVX+E. Sham vs OVX: p=0.1386, OVX vs OVX+E: p=0.9991, Sham vs OVX+E: p=0.08120. The upper, middle, and lower bars show the third quartile, median, and the first quartile, respectively. Steel–Dwass test, *p<0.05. n.s., not significant. (**C**) Photographs of

*Figure 5 continued on next page*

*Figure 5 continued*

brain slices after experiments of double in situ hybridization (*agrp1* [green] and *estrogen receptors* [magenta]). (**D**) Expanded photograph of *agrp1* and *esr2a* co-expressing neurons (white arrowhead). Scale bars: 50 µm.

The online version of this article includes the following source data and figure supplement(s) for figure 5:

**Source data 1.** The numerical data for *Figure 5*.

**Figure supplement 1.** Food intake of OVX female does not show significant change according to the day-length.

**Figure supplement 1—source data 1.** The numerical data for *Figure 5—figure supplement 1*.

## Discussion

In the present study, we took advantage of female medaka, which clearly shows breeding season-dependent feeding behavior and found that neuropeptides, *agrp1* and *npyb,* show higher expression under the breeding condition than under the non-breeding condition. We also obtained results to suggest that the expression of both *agrp1* and *npyb* changes depending on nutritional status of female medaka. In addition, ovariectomy and E2 administration changed expression of *agrp1* but not *npyb*, suggesting that increased release of ovarian E2 in the breeding season upregulates the *agrp1* expression, which results in the facilitation of female feeding behavior. Finally, loss-of-function mutation of AgRP1 decreased the amount of food intake and the number of spawned eggs. The present results suggest that breeding season-dependent feeding behavior can be modulated by the increased expression of *agrp1* upregulated by increased release of ovarian estrogen in the breeding season (*Figure 7*). To date, not a few previous reports have shown the influence of nutritional status on reproduction (*Evans and Anderson, 2012*; *Kalra and Kalra, 1996*; *Kirkwood et al., 1987*; *Merry and Holehan, 1979*; *Hasebe et al., 2016*). On the other hand, although seasonal breeders have been reported to show changes in feeding behavior during the breeding season (*Tsuyuki, 2018*; *Kawai et al., 2020*), its neuroendocrine mechanisms have largely remained enigmatic. Our present results may provide a neuroendocrinological model for the mechanisms that play a key role in the control of breeding season-dependent feeding behavior in teleosts.

### Feeding-related peptides AgRP and NPY in medaka

Here, we demonstrated that female medaka eat more under the condition of breeding season (*Figure 1B*). Various kinds of neuropeptides in the brain have been suggested to control feeding, and these are generally called 'feeding-related peptides' (*Funahashi et al., 2003*). In the present study, we first used a seasonally breeding model teleost medaka and searched for the 'feeding-related peptides' involved in seasonal feeding behavior. A whole-brain RNA-seq analysis using female medaka under the breeding condition (LD) and non-breeding condition (SD) revealed two kinds of feeding-related peptides, *agrp1* and *npyb*, which show different expression levels between LD and SD (*Figure 1—figure supplement 2*). In mammals, AgRP and NPY are known to have orexigenic function and are co-expressed in hypothalamic neurons (*Schwartz et al., 2000*; *Hahn et al., 1998*). Previous studies in mammals (*Schwartz et al., 2000*; *Andermann and Lowell, 2017*; *Muroi and Ishii, 2016*) have suggested neural mechanisms of appetite including functions of AgRP and NPY. However, such mechanisms in non-mammalian vertebrates such as teleosts (*Rønnestad et al., 2017*; *Blanco and Soengas, 2021*) have not yet been clarified. Our present study using medaka has shown possible functions of AgRP and NPY in teleost feeding behavior, especially in a breeding season-dependent manner.

Our present study using medaka showed that female medaka express *agrp1* in hypothalamus, and food restriction increases the *agrp1* expression (*Figure 2B and G*). It has been reported that leptin receptor-knockout medaka show higher food intake and higher expressions of *agrp1* and *npya* than wild type, whereas the expression of *agrp2* and *npyb* remained to be analyzed (*Chisada et al., 2014*). Zebrafish has also been used as a model animal in teleosts. In zebrafish, food restriction increased *agrp1* (*Song et al., 2003*; *Opazo et al., 2018*) expression, and transgenic overexpression of *agrp1* caused gain of body weight (*Song and Cone, 2007*), as in mammals (*Graham et al., 1997*; *Adam et al., 2002*; *Hahn et al., 1998*; *Ilnytska and Argyropoulos, 2008*). It has also been reported that *agrp1* knockout zebrafish eat less than the wild type (*Shainer et al., 2019*), although loss of AgRP in mice showed little effect on food intake (*Qian et al., 2002*). Our present results and these previous studies strongly support that *agrp1* regulates feeding and may act as an orexigenic factor in teleosts. On the other hand, *agrp2* neurons were distributed mainly in telencephalon, and its weak expressions

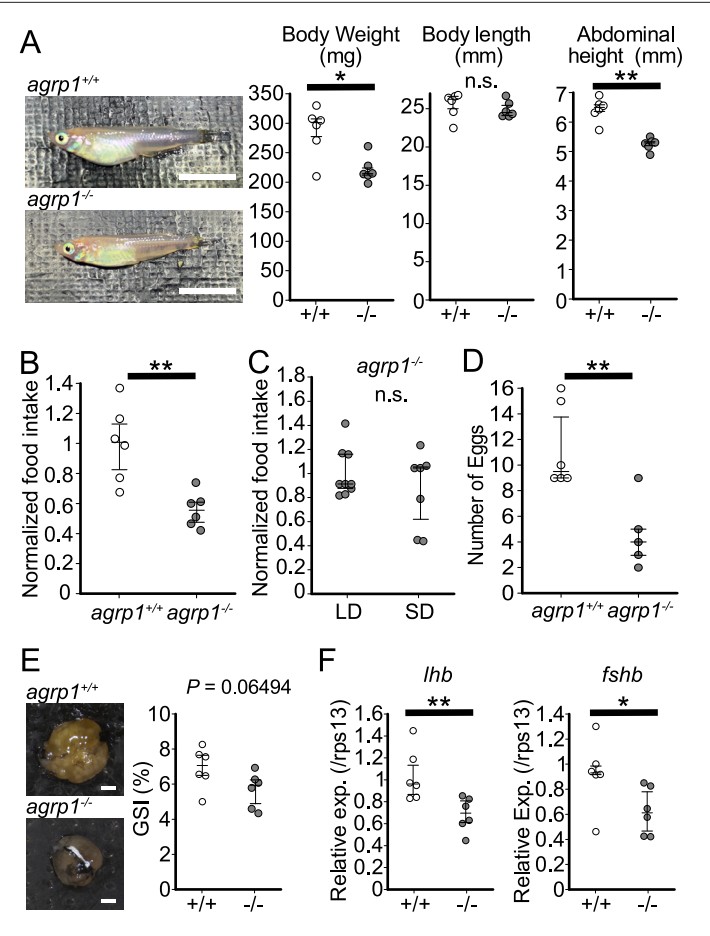

**Figure 6.** *agrp1⁻ᐟ⁻* female medaka show decrease in food intake and the number of fertilized eggs. (**A**) Lateral views of representative *agrp1⁺ᐟ⁺* and *agrp1⁻ᐟ⁻* female medaka (left), and body weight, body length, and abdominal height (right) of *agrp1⁺ᐟ⁺* (white; n=6) and *agrp1⁻ᐟ⁻* (gray; n=6). All the fish are 4-month-old adult medaka. Scale bar: 1 cm. body weight: p=0.04113, *U*=31; body length: p=0.3939, *U*=24; abdominal height: p=0.002165, *U*=36. (**B**) Food intake (10 min) of *agrp1⁺ᐟ⁺* (white; n=6) and *agrp1⁻ᐟ⁻* (gray; n=6) female medaka. Each amount of food intake is normalized by the average number of that of *agrp1⁺ᐟ⁺* medaka (p=0.004329, *U*=35). (**C**) Food intake (10 min) of LD *agrp1⁻ᐟ⁻* (white; n=9) and SD *agrp1⁻ᐟ⁻* (gray; n=7) female medaka. Each amount of food intake is normalized by the average number of that of LD *agrp1⁻ᐟ⁻* medaka (*p*=0.5953, *U*=37). (**D**) The number of eggs spawned by *agrp1⁺ᐟ⁺* (white; n=6) and *agrp1⁻ᐟ⁻* (gray; n=5) female medaka in a day. Each female was paired with a wildtype male (p=0.008658, *U*=28.5). (**E**) Photograph of ovary in representative *agrp1⁺ᐟ⁺* and *agrp1⁻ᐟ⁻* female (left) and the gonado-somatic index (GSI, right). n=6 of each group (p=0.06494, *U*=30). Scale bar: 1 mm. (**F**) Expression of gonadotropin genes (*lhb* and *fshb*) in the pituitary of *agrp1⁺ᐟ⁺* (white; n=6) and *agrp1⁻ᐟ⁻* (gray; n=6) female medaka. *lhb*: (p=0.008658, *U*=34); *fshb*: (p=0.02597, *U*=32). The upper, middle, and lower bars show the third quartile, median, and the first quartile, respectively. Mann–Whitney *U* test, *p<0.05, **p<0.01. n.s., not significant.

The online version of this article includes the following source data and figure supplement(s) for figure 6:

**Source data 1.** The numerical data for *Figure 6*.

**Figure supplement 1.** Mutation site of *agrp1* knockout medaka.

**Figure supplement 2.** Food intake of WT, *agrp1* hetero, and homo knockout medaka.

**Figure supplement 2—source data 1.** The numerical data for *Figure 6—figure supplement 2*.

were also observed in the POA and the hypothalamus (*Figure 2C*), which is different from the results in zebrafish (*Shainer et al., 2017*). In the present study, food restriction did not remarkably affect the *agrp2* expression in medaka (*Figure 2*, *Figure 2—figure supplement 1E*), and AgRP2 in zebrafish is suggested to play an important role in stress response, not feeding (*Shainer et al., 2019*). Thus, it is highly probable that *agrp2* is involved in functions other than feeding in female medaka.

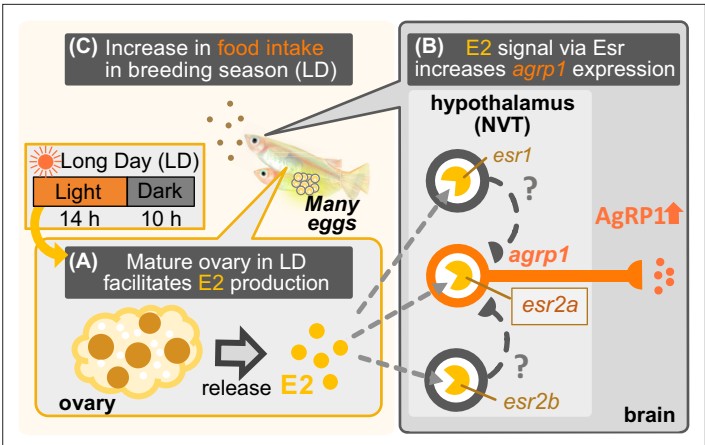

**Figure 7.** Illustration of mechanisms of breeding season-dependent feeding behavior in medaka suggested by the present study. (**A**) Long day (LD) condition in the breeding season induces ovarian maturation, which facilitates release of estrogen (E2) from the mature ovary. (**B**) High concentration of serum E2 increases *agrp1* expression in the brain via the estrogen receptors, especially, *esr2a*. (**C**) Higher expression of AgRP1 is suggested to activate neural circuitry for feeding, which leads to an increase in food intake and egg spawning of female medaka in the breeding season.

Furthermore, we demonstrated that *npya* is expressed in multiple brain regions including hypothalamus in medaka (***Figure 2D***), which is similar to mammals (***Gray and Morley, 1986***) and zebrafish (***Yokobori et al., 2012***; ***Jeong et al., 2018***). We also showed that *npya* is not co-expressed with *agrp1* nor *agrp2* (***Figure 2F***) as in the zebrafish (***Jeong et al., 2018***), suggesting that the relationship between NPY and AgRP of teleosts may be different from that of mammals, in which most of the *agrp*-expressing neurons co-express *npy* (***Hahn et al., 1998***). Moreover, the present study also showed that *npyb* expression is localized in telencephalon (***Figure 2E***), which is similar to the previous report using tiger puffer (***Kamijo et al., 2011***). Previous studies of NPY in zebrafish showed that zebrafish has only one type of NPY (NPYa) (***Söderberg et al., 2000***; ***Larsson et al., 2009***) and has lost NPYb during evolution. Like in mammals (***Marks et al., 1992***; ***Clark et al., 1984***; ***Glenn Stanley et al., 1986***; ***Marks and Waite, 1997***; ***Baldock et al., 2009***), food restriction in zebrafish increased the *npya* expression in the hypothalamus (***Song et al., 2003***; ***Opazo et al., 2018***), and intracerebroventricular administration of NPYa increased food intake (***Yokobori et al., 2012***). Although these zebrafish studies suggest that NPYa may increase food intake, it is still debatable since body weight was not significantly different between *npya* knockout zebrafish and wild type (***Shiozaki et al., 2020***). Interestingly, by analyzing both *npya* and *npyb* expression in medaka of different nutritional conditions, we found that food restriction decreased the *npya*-expressing cell number in the hypothalamus (***Figure 2—figure supplement 1***) and *npyb* expression level (***Figure 2J***). These changes in *npya* and *npyb* expressions are not consistent with previous studies using other conventional model animals described above (***Yokobori et al., 2012***; ***Marks et al., 1992***; ***Clark et al., 1984***; ***Glenn Stanley et al., 1986***; ***Marks and Waite, 1997***; ***Baldock et al., 2009***). The present study may suggest that the function of *npy* may be different among teleosts. In addition, *npyb* expression was increased under the breeding condition (LD), while LD female showed increase in food intake (***Figure 3F***). Thus, decreased expression of *npyb* by food restriction (***Figure 2J***) may suggest that the change in *npyb* expression reflects nutritional condition in medaka. Thus, future study of *npya* and *npyb* functions in the control of feeding will be necessary.

As described above, we found that *agrp1*, *npya*, and *npyb* change expression levels in response to nutritional status. Among these three genes, we suggest that *agrp1* most probably affects relatively long-term feeding in the breeding season, which agrees well with the recent studies in mice showing the function of AgRP as a long-term orexigenic factor. In mice, it has been reported that intracerebroventricular administration of AgRP increases food intake for 1 week (***Hagan et al., 2000***), and stimulation of receptors expressed in AgRP neurons triggers AgRP release, leading to an increase in food intake for 3 days (***Nakajima et al., 2016***). The present study also suggests a long-term (seasonal) orexigenic effect of AgRP in teleosts and may also provide an important insight into the understanding of common regulatory mechanisms of feeding by AgRP among various animal species.

## High concentration of E2 in the breeding season facilitates *agrp1* expression

Our results suggest that *agrp1* and *npyb* show higher expressions under the breeding condition (LD) (*Figure 3*), but the experiments using juvenile female medaka (*Figure 4*) showed that expression levels of these two genes do not change according to the day-length itself but to the LD-induced sexual maturity. In addition, the present results indicate that the ovarian estrogen E2 upregulates *agrp1* expression mainly via the estrogen receptors *esr2a* that are co-expressed in some population of *agrp1* neurons in the hypothalamic nucleus NVT (*Figure 5*). Since LD female medaka (breeding) shows high blood concentration of E2 (*Ikegami et al., 2022*), this pathway may be important for breeding season-dependent feeding behavior. Especially, in teleost, main egg protein for nutrition is vitellogenin, whose expression is also facilitated by E2 (*Tohyama et al., 2017*). Taken together, it is suggested that E2 may synchronously regulate amount of food intake and female-specific reproductive signals (vitellogenin production and oocyte maturation), which plays a key role in reproductive success in oviparous animals.

In mammals, previous studies have reported on inconsistent effects of ovary and E2 on feeding. Ablation of ovary caused suppression of food intake in mice (*Yu et al., 2020*), whereas it caused no change in rats (*Roesch, 2006*). On the other hand, administration of E2 decreased food intake in both mice (*Yu et al., 2020*) and rats (*Roesch, 2006*). It should be noted that these laboratory rodents only exhibit short estrous cyclicity and have lost breeding seasonality, and the blood E2 concentration drastically changes in a few days (*Nilsson et al., 2015*). Thus, it is possible that the control mechanisms of feeding may be different between animals with short estrous cyclicity and those with breeding seasonality.

Furthermore, the present study suggests different control mechanisms of feeding between the animals with breeding seasonally and those without. Here, we showed that E2 directly modulates *agrp1* expression via *esr2a* receptors co-expressed in the *agrp1* neurons (*Figure 5C*), while in mice, *AgRP/NPY* neurons are reported to be suppressed by E2 indirectly via *esr1* (*erα*)-expressing *Kiss1* neurons located in the hypothalamic arcuate nucleus (*Qiu et al., 2018*; *Dubois et al., 2016*; *Yang et al., 2017*). On the other hand, in medaka, expressions of all kinds of estrogen receptors are reported to be localized in NVT (*Zempo et al., 2013*), in which *agrp1* expression is also localized (*Figure 2B*). In addition, *esr2a* has been reported to be involved in the feedback regulation of follicle-stimulating hormone in the pituitary and in the development of oviduct, and *esr2a* knockout females are completely infertile (*Kayo et al., 2019*). Our hypothesis that estrogen signaling via *esr2a* affects *agrp1* expression may highlight another important function of *esr2a* for reproduction, while a possibility still remains that *esr1*- and *esr2b*-expressing neurons also affect *agrp1* expression indirectly.

## AgRP1 changes feeding behavior depending on LD-induced sexual maturity, which causes the increase in food intake in the breeding season

Since our results thus far indicate the importance of *agrp1*, which shows upregulated expression directly stimulated by the ovarian E2 in the breeding season, we examined phenotypes of *agrp1* knockout (*agrp1*$^{-/-}$) medaka. We found that *agrp1*$^{-/-}$ medaka under the condition of breeding season eat less (*Figure 6B*) and spawn a smaller number of eggs (*Figure 6D*) than WT. Furthermore, the *agrp1*$^{-/-}$ females did not show significant difference of food intake in LD and SD (*Figure 6C*). These results strengthen our hypothesis that *agrp1* is involved in the increased food intake in the breeding season. Furthermore, *agrp1*$^{-/-}$ female displayed light body weight (*Figure 6A*), accompanied by smaller ovary (*Figure 6E*) and low level of expression of gonadotropins, *fshb* and *lhb* (*Figure 6F*), which are considered to have caused smaller number of spawned eggs. All of these results support our hypothesis that AgRP1 plays an important role in the breeding season-dependent feeding behavior, which culminates in normal reproduction.

In summary, by using a seasonal breeder medaka, we found evidence to suggest that long day-length facilitates ovarian maturation and E2 release, which upregulates *agrp1* expression of hypothalamic neurons to activate neural circuitry for feeding behavior and boost oocyte maturation. We propose that this kind of positive feedback control may be important for animals that spawn many eggs every day in the breeding season (*Figure 7*); medaka needs plenty of food for the production of many eggs. In other words, the metabolic costs of producing eggs on a daily basis in medaka would

inevitably require increased food intake. Indeed, previous study showed a need for high food intake for reproduction (*Hasebe et al., 2016*). It is expected that future studies will elucidate whether or not the present findings in medaka are applicable to other seasonal breeders as well.

## Materials and methods

### Animals

Female and male wild-type d-rR medaka (*O. latipes*) and *agrp1* knockout (*agrp1$^{-/-}$*) medaka were maintained in pairs or shoals at 27°C. Fish were fed three times a day with brine shrimp and flake food (Otohime B-2; San-u Fish Farm, Osaka, Japan). Their reproductive status was controlled by day-length (LD [14 h light/10 h dark]: reproductive, SD [10 h light/14 h dark]: non-reproductive). The light-on time was 8:00 AM. We used juvenile medaka (~5 weeks after fertilization) and adult medaka (>3 months after fertilization). Female medaka, which spawned at least three consecutive days, were used as reproductive ones. For the analysis of the effect of food restriction, reproductive and non-reproductive female medaka were fasted for 14 days or 10 days. Note that all medaka survived after food restriction. For the analyses of food intake, we food-restricted medaka for 6 h after 10 min feeding in the morning and sampled their whole brains for subsequent experiments. Food-restricted medaka were sampled at the same time as the other fed medaka. All experiments and fish maintenance were conducted in accordance with the Guidelines for Proper Conduct of Animal Experiments (Science Council of Japan) and the protocols approved by the Animal Care and Use Committee of Graduate School of Science, the University of Tokyo (permission number, 17-1, 20-6), and the Animal Care and Use Committee of Graduate School of Agriculture, Tokyo University of Agriculture and Technology (permission number, R05-15, R06-27).

### Food intake assay

Each 6 h food-restricted medaka was put into a white cup with 100 mL breeding water and was habituated for 5 min. Then, we fed medaka by application of 200 µL aliquots of food water containing brine shrimp in all-you-can-eat style and serve another aliquot once done with it, which is repeated N times (like the Japanese 'Wanko soba'; so-called Japanese 'Wanko soba' method). Then, 10 min after the start, we stopped feeding medaka and placed a magnetic bar to stir the breeding water so that the shrimp concentration will be constant. Then, we collected 10 mL aliquot from the experimental cup by using a micro pipette and transferred it to a conical tube. The conical tube was frozen overnight, and the leftover brine shrimp sunk in the bottom were counted by 'shrimp-counter'. We counted the number of brine shrimp in the 200 µL solution three times before and after the experiments, and the average number was used. The food intake was calculated as follows.

> (Food Intake) = (The average number of brine shrimp in the solution) * (number of aliquots, N) - (number of leftover brine shrimp sunk in the bottom) * 10

Food intake was normalized by the average of LD or WT medaka.

### 'Shrimp-counter' system

The number of shrimps in the solution was counted using OpenCV3 library (Intel, Santa Clara, CA) run under a Python script (*Figure 1—source code 1*). This script was run under Anaconda 4.4.0 for Windows running Python 3.5.

### RNA-sequencing

We collected two whole brains of LD or SD female in one tube (note that pituitary was confirmed not to be included) and extracted total RNA by using NucleoSpin RNA Plus kit (MACHEREY-NAGEL, Düren, Germany). cDNA was obtained by KAPA Stranded mRNA-Seq Kit (Kapa Biosystems, Inc, Wilmington, MA) and KAPA Library preparation kit (Kapa Biosystems, Inc). Then, it was applied to a next-generation sequencer Hiseq 2500 (Illumina, San Diego, CA), following the standard protocol of Illumina system. We selected the candidate genes judging from transcripts per million (TPM) for expression value in the obtained data using CLC Genomics Workbench. We made volcano plot using R (*R Development Core Team, 2023*) and RStudio (2023) and colored dots, which indicate p-value

<0.05 and |log FC|>1. In addition, we made a heatmap of genes related to neuroendocrine system using DESeq2 (*Love et al., 2014*).

## Histological analysis of the distribution of *agrp*- and *npy*-neurons in the brain

To analyze the distribution of *agrp*- and *npy*-expressing neurons, we performed ISH for *agrp* and *npy* on frozen sections of reproductive medaka. In brief, female medaka was anesthetized (FA100, Bussan Animal Health Co, Ltd, Osaka, Japan), and its brain was picked up and fixed with 4% para-formaldehyde (PFA)/PBS. In analyses on *agrp2* expression, we performed perfusion-fixation by using 4% PFA/PBS. After incubation with 30% sucrose/PBS, brains were embedded in 5% low melting agar/20% sucrose/PBS and sectioned at a thickness of 25 μm. The sections were hybridized with *agrp1* (ENSORLG00000000398, 177 bases), *agrp2* (ENSORLG00000029106, 303 bases), *npya* (ENSORLG00000004649, 288 bases) and *npyb* (ENSORLG00000007880, 288 bases)-specific digoxigenin (DIG)-labeled RNA probes and performed nitro blue tetrazolium (NBT)/ 5-bromo-4-chloro-3-indolyl-phosphate color-reaction (BCIP) after wash and incubation with anti-digoxigenin antibody (Cat# 11093274910; Roche; RRID:AB_514497) as previously reported (*Zempo et al., 2013*). Photographs were taken with a digital camera (DFC310FX; Leica Microsystems, Wetzlar, Germany) attached to an upright microscope (DM5000B; Leica Microsystems).

## Histological analysis of *agrp1*-, *npy*-, and estrogen receptor (*esr*)-expressing neurons

To examine whether *agrp1*-expressing neurons co-express *npya* and *esr*, we prepared *agrp1* fluorescein-labeled RNA probe and carried out double ISH as previously reported (*Umatani et al., 2022*). *esr* DIG-labeled probes were kindly given by Dr. Kayo (Kyoto Univ.), and we used *npya* DIG-labeled probe described in the previous paragraph. In brief, we made brain sections as described above and applied both *agrp1* fluorescein-labeled and each DIG-labeled RNA probes. Signals for *npya*, *esr1*, *esr2a*, and *esr2b* were visualized by incubation with anti-digoxigenin antibody (Cat# 11207733910; Roche; RRID:AB_514500) and TSA Plus Cy3 System (TSA-Plus Cyanine 3 system, Cat# NEL744001KT, Akoya Biosciences, Marlborough, MA). After inactivation of Cy3 system by 3% $H_2O_2$, we applied peroxidase-conjugated anti-fluorescein antibody (Cat# 11426346910, Roche; RRID:AB_840257) on sections and performed TSA Plus biotin system (Cat# NEL749A001KT, Akoya Biosciences). Then, signals for *agrp1* were visualized by Alexa 488 conjugated streptavidin (Cat# S11223, Invitrogen). For counter-staining of cell nuclei, DAPI in PBS was applied on section. On the other hand, to examine whether *agrp2*-expressing neurons co-express *npya*, we used *npya* fluorescein-labeled RNA probe and *agrp2* DIG-labeled one. Double ISH of them was performed according to the same method described above. Fluorescent images were acquired with a confocal laser-scanning microscope (AXR, Nikon, Tokyo, Japan) using excitation and emission wavelengths of 405 nm and 429–474 nm for DAPI, 488 nm and 512–526 nm for Alexa 488, and 561 nm and 571–625 nm for Cy3, respectively. These were photographed at the Tokyo University of Agriculture and Technology for Smart Core facility Promotion Organization.

## Quantitative real-time polymerase chain reaction (RT-qPCR)

A whole brain or a pituitary was collected from each medaka and total RNA was extracted by using FastGene RNA basic kit (Nippon Genetics Co, Ltd) according to the manufacturer's instructions. For the juvenile medaka, we checked their sex as previously reported, and two samples of the same sex were mixed and used as one sample. Total RNA samples were reverse transcribed by FastGene cDNA synthesis 5×ReadyMix OdT according to the manufacturer's instructions. For the analyses of the brain, 1 μL of cDNA diluted with 10-fold MQ was mixed with KAPA SYBR Fast qPCR kit (Kapa Biosystems, Inc) and amplified with Lightcycler96 [Roche; 95°C 150 s (95°C 10 s, 60°C 10 s, 72°C 15 s)×45 cycles]. For the analysis of the pituitary, 1 μL of cDNA diluted with fivefold MQ was mixed with KAPA SYBR Fast qPCR kit and amplified with Lightcycler96 [Roche; 95°C 150 s (95°C 10 s, 60°C 10 s, 72°C 10 s)×45 cycles]. The data was normalized by housekeeping gene, ribosomal protein s13 (*rps13*). Primer sequences were as follows:

AgRP1 RT-PCR F1 CCAATTTCCAGTCACCGAAG

AgRP1 RT-PCR R1 CTGGGTCCAACACAGAATCA
AgRP2 RT-PCR F1 TTGTTGTGCTTCTTGCTGCT
AgRP2 RT-PCR R1 ACAGAGCTCCAAACGGTGTC
NPYa SE CTCATCACAAGACAGAGGTATGGG
NPYa AS GGGTTGTAACTTGACTGTGGAAGTG
NPYb SE CTGCCTGCTCCTCTGTTTTTTCTC
NPYb AS CACAGTGTCTGGGTTGTCTCTCTTTC
qPCR FSHb Fw new TGGAGATCTACAGGCGTCGGTAC
qPCR FSHb Rv new AGCTCTCCACAGGGATGCTG
qPCR LHb Fw new AGGGTATGTGACTGACGGATCCAC
qPCR LHb Rv new TGCCTTACCAAGGACCCCTTGATG
RPS13 SE GTGTTCCCACTTGGCTCAAGC
RPS13 AS CACCAATTTGAGAGGGAGTGAGAC

## Sham operations, ovariectomy, and E2 administration

Ovariectomy and E2 administration were performed according to a previous study (*Kayo et al., 2020*). Briefly, reproductive female medaka were anesthetized with 0.02% MS-222 (Sigma-Aldrich, St. Louis, MO) and their ovaries were excised via intraperitoneal operation. Sham operation group was anesthetized, received an abdominal incision without removing the ovaries, and received skin suture by using a silk thread. After checking that all Sham females spawn, we prepared three tanks; two tanks contained 7–8 OVX medaka, and one tank contained Sham medaka in 2 L breeding water in it. We dissolved β-estradiol 1.4 mg in 1 mL EtOH (E2 stock) and dispensed 2 µL of E2 stock or the same amount of 100% ethanol for the control tank. Ethanol or E2-containing water were changed every day. After the steroid treatment for 5 days, the medaka were anesthetized, and their whole brains were collected for RT-qPCR analysis.

## Generation of *agrp1* KO medaka lines

We generated *agrp1* KO medaka lines by using CRISPR/Cas9. Cas9 mRNA and tracer RNA were purchased from Integrated DNA Technologies (IDT, Coralville, IA). The guide RNA sequence for digestion by CRISPR/Cas9 complex was 'CCTCACCAGCAGTCCTGCCTGG'.

Mixture of Cas9 protein, tracer RNA, CRISPR RNA, GFP mRNA diluted with PBS and 0.02% phenol red (final concentration: Cas9 protein; 500 ng/µL, tracer RNA; 100 ng/µL, CRISPR RNA; 50 ng/µL, GFP mRNA; 5 ng/µL) was injected into the cytoplasm of one- or two-cell-stage embryos (F0). To obtain homozygous transgenic offspring, the carriers were crossed with each other.

## Measurement of body size and ovary size

We took photographs of fish bodies from the lateral side by using a digital camera MC120HD (Leica) and calculated the abdominal and body length using ImageJ. For the analysis of body length, we measured the length from the mouth to the base of the tail. GSI was calculated as ovary weight/ body weight * 100.

## Immunohistochemistry using AgRP1 antibody

We made brain sections of WT and *agrp1* KO as described in the 'Histological analysis of the distribution of agrp- and npy-neurons in the brain'. After washing with PBST two times for 10 min, sections were incubated with AgRP1 antibody (1:1000; rabbit polyclonal AgRP [83-132] amide [human], Cat# H-003-53; RRID:AB_2313908, Phoenix Pharmaceutical, Burlingame, CA)/0% goat serum/PBS overnight. On the next day, slides were washed with PBST and incubated with anti-rabbit biotinylated goat antibody (1:200; Cat#BA-1000; Vector Laboratories, Burlingame, CA) for 1 h. Then we applied Alexa 488 conjugated streptavidin (1:500, Cat#S11223, Invitrogen) and DAPI (1:2000; Dojindo Laboratories, Kumamoto, Japan). Photographs were taken with a digital camera (Leica Microsystems) attached to an upright microscope (Leica Microsystems).

## Statistics

For the statistical analysis, we used Kyplot 5.0 software (Kyence, Osaka, Japan) or R software (*R Development Core Team, 2023*) with RStudio (version 2023.06.0+421). For the comparison of the TPM,

we used Student's *t*-test. For the comparison of *agrp1* and *npyb* expressions in OVX and E2-administrated medaka, Steel–Dwass test was used for multiple comparison of the expression level. In the other experiments, we used Mann–Whitney *U* test. In all statistical analysis, significance levels were described as follows: *p<0.05, **p<0.01, and ***p<0.001.

## Code availability

The code of 'Shrimp-counter' system is available in the present study in *Figure 1—source code 1*.

## Acknowledgements

We thank Drs. Mikoto Nakajo (Osaka Med and Pharm Univ), Soma Tomihara (Hiroshima Univ), and Kana Ikegami (Kitasato Univ) for helpful discussion. We also thank Dr. Daichi Kayo (Kyoto Univ.) for his kind supply of *esr* DIG-labeled probes. In addition, we deeply appreciate Dr. Hiroyuki Takeda (U Tokyo) for his continued kind support and encouragement during our experiments. We also thank Dr. Yutaka Miura (Tokyo Univ of Agri and Tech) for his kind support of our experiments. As for the help of animal care, we thank Ms. Hisako Kohno, Miho Kyokuwa, Hiroko Tsukamoto, and Risa Nakaba. We thank Ms. Maiko Matsuda (Vesper studio) for her kind gift of medaka illustrations to us. The present study was supported by JSPS KAKENHI (JP21J20864 and JP22KJ0597 to YT, JP26221104 and JP21K06262 to YO, JP20H03071 to CU), and MEXT Initiative for Realizing Diversity in the Research Environment (Leadership training for women) (CU).

## Additional information

### Funding

| Funder | Grant reference number | Author |
|---|---|---|
| Japan Society for the Promotion of Science | JP21J20864 | Yurika Tagui |
| Japan Society for the Promotion of Science | JP26221104 | Yoshitaka Oka |
| Japan Society for the Promotion of Science | JP20H03071 | Chie Umatani |
| Ministry of Education, Culture, Sports, Science and Technology | MEXT Initiative for Realizing Diversity in the Research Environment | Chie Umatani |
| Japan Society for the Promotion of Science | JP22KJ0597 | Yurika Tagui |
| Japan Society for the Promotion of Science | JP21K06262 | Yoshitaka Oka |

The funders had no role in study design, data collection and interpretation, or the decision to submit the work for publication.

### Author contributions

Yurika Tagui, Conceptualization, Data curation, Funding acquisition, Validation, Investigation, Visualization, Methodology, Writing - original draft, Writing – review and editing; Shingo Takeda, Hiroyo Waida, Data curation, Validation, Investigation, Methodology; Shoichi Kitahara, Data curation, Validation, Investigation, Visualization; Tomoki Kimura, Software, Methodology; Shinji Kanda, Software, Supervision, Funding acquisition, Investigation, Methodology, Writing – review and editing; Yoshitaka Oka, Yu Hayashi, Resources, Supervision, Funding acquisition, Writing – review and editing; Chie Umatani, Conceptualization, Resources, Data curation, Formal analysis, Supervision, Funding acquisition, Validation, Investigation, Visualization, Methodology, Writing - original draft, Project administration, Writing – review and editing

### Author ORCIDs

Shinji Kanda ⓘ https://orcid.org/0000-0001-7556-1433

Yoshitaka Oka [ID] https://orcid.org/0000-0002-3482-3051
Yu Hayashi [ID] https://orcid.org/0000-0001-5007-4532
Chie Umatani [ID] https://orcid.org/0000-0003-3612-4374

### Ethics

All experiments and fish maintenance were conducted in accordance with the Guidelines for Proper Conduct of Animal Experiments (Science Council of Japan) and the protocols approved by the Animal Care and Use Committee of Graduate School of Science, the University of Tokyo (Permission number, 17-1, 20-6) and the Animal Care and Use Committee of Graduate School of Agriculture, Tokyo University of Agriculture and Technology (Permission number, R05-15, R06-27).

Reviewer #1 (Public review): https://doi.org/10.7554/eLife.100996.3.sa1
Reviewer #2 (Public review): https://doi.org/10.7554/eLife.100996.3.sa2
Reviewer #3 (Public review): https://doi.org/10.7554/eLife.100996.3.sa3
Author response https://doi.org/10.7554/eLife.100996.3.sa4

## Additional files

### Supplementary files
MDAR checklist

### Data availability

All data generated or analyzed during this study are included in the manuscript and supporting files; source data files have been provided for Figures 1–6.

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
