## [Editor Report · eLife Assessment]

This article provides **fundamental** new insight into the mechanisms linking photoperiod, reproduction function, and feeding activity, using medaka, a genetic model that itself exhibits photoperiodic responses. As well as identifying key neuropeptide genes that are regulated by photoperiod and involved in regulating feeding activity, the authors establish a knockout line for agrp1 using CRISPR Cas9-based approach, profiting from the extensive use and development on this methodology in medaka. The combination of the RNAseq and quantitative in situ hybridization analysis with the knockout results as well as the study of ovariectomized fish provides **compelling** evidence implicating agrp1 in feeding regulation in response to photoperiod and reproductive status.

---

## [Referee Report · Reviewer #1 (Public review)]

Summary:

The authors use the teleost medaka as an animal model to study the effect of seasonal changes in day-length on feeding behaviour and oocyte production. They report a careful analysis how day-length affects female medakas and a thorough molecular genetic analysis of genes potentially involved in this process. They show a detailed analysis of two genes and include a mutant analysis of one gene to support their conclusions

Strengths:

The authors pick their animal model well and exploit the possibilities to examine in this laboratory model the effect of a key environmental influence, namely the seasonal changes of day-length. The phenotypic changes are carefully analysed and well controlled. The mutational analysis of the agrp1 by a ko-mutant provides important evidence to support the conclusions. Thus this report exceeds previous findings on the function of agrp1 and npyb as regulators of food-intake and shows how in medaka these genes are involved in regulating the organismal response to an environmental change. It thus furthers our understanding on how animals react to key exogenous stimuli for adaptation.

Weaknesses:

The authors are too modest when it comes to underscoring the importance of their findings. Previous animal models used to study the effect of these neuropeptides on feeding behaviour have either lost or were most likely never sensitive to seasonal changes of day-length. Considering the key importance of this parameter on many aspects of plant and animal life it could be better emphasised that a suitable animal model is at hand that permits this.

The molecular characterization of the agrp1 ko-mutant that the authors have generated lacks some details that would help to appreciate the validity of the mutant phenotype. Additional data would help in this respect.

Comments on revisions:

The authors dealt adequately with the comments and suggestions of this reviewer.

---

## [Referee Report · Reviewer #2 (Public review)]

Summary:

The authors investigated the mechanisms behind breeding season-dependent feeding behavior using medaka, a well-known photoperiodic species, as a model. Through a combination of molecular, cellular, and behavioral analyses, including tests with mutants, they concluded that AgRP1 plays a central role in feeding behavior, mediated by ovarian estrogenic signals.

Strengths:

This study offers valuable insights into the neuroendocrine mechanisms that govern breeding season-dependent feeding behavior in medaka. The multidisciplinary approach, which includes molecular and physiological analyses, enhances the scientific contribution of the research.

Comments on revised version:

My concerns from the first review have been addressed. The manuscript's key points are clearly presented, and the conclusions are readily comprehensible

---

## [Referee Report · Reviewer #3 (Public review)]

Summary:

Understanding the mechanisms whereby animals restrict the timing of their reproduction according to day length is a critical challenge given that many of the most relevant species for agriculture are strongly photoperiodic. However, the principal animal models capable of detailed genetic analysis do not respond to photoperiod so this has inevitably limited progress in this field. The fish model medaka occupies a uniquely powerful position since it's reproduction is strictly restricted to long days and it also offers a wide range of genetic tools for exploring, in depth, various molecular and cellular control mechanisms.

For these reasons, this manuscript by Tagui and colleagues is particularly valuable. It uses the medaka to explore links bridging photoperiod, feeding behaviour and reproduction. The authors demonstrate that in female, but not male medaka, photoperiod-induced reproduction is associated with an increase in feeding, presumably explained by the high metabolic cost of producing eggs on a daily basis during the reproductive period. Using RNAseq analysis of the brain, they reveal that the expression of the neuropeptides agrp and npy that have been previously implicated in the regulation of feeding behaviour in mice, are upregulated in the medaka brain during exposure to long photoperiod conditions. Unlike the situation in mouse, these two neuropeptides are not coexpressed in medaka neurons and food deprivation in medaka led to increases in agrp but also a decrease in npy expression. Furthermore, the situation in fish may be more complicated than in mouse due to the presence of multiple gene paralogs for each neuropeptide. Exposure to long day conditions increases agrp1 expression in medaka as the result of increases in the number of neurons expressing this neuropeptide, while the increase in npyb levels results from increased levels of expression in the same population of cells. Using ovariectomized medaka and in situ hybridization assays, the authors reveal that the regulation of agrp1 involves estrogen acting via the estrogen receptor esr2a. Finally, a loss of agrp1 function mutant is generated where the female mutants fail to show the characteristic increase in feeding associated with long day enhanced reproduction as well as yielding reduced numbers of eggs during spawning.

Strengths:

This manuscript provides important foundational work for future investigations aiming to elucidate the coordination of photoperiod sensing, feeding activity and reproduction function. The authors have used a combination of approaches with a genetic model that is particularly well suited to studying photoperiodic dependent physiology and behaviour. The data are clear and the results are convincing and support the main conclusions drawn. The findings are relevant not only for understanding photopriodic responses but also provide more general insight into links between reproduction and feeding behaviour control.

The manuscript has been further strengthened by the inclusion of additional information according to my advice: The analysis of ovariectomized female fish and juvenille fish has now been reported in terms of their feeding behaviour and so provide a complete view of the position of this feeding regulatory mechanism in the context of reproduction status. Furthermore, the discussion section has been expanded to speculate on the functional significance of linking feeding behaviour control with reproductive function. Modifications made in order to address technical concerns of the other 2 reviewers have also significantly strengthened the presentation of this work.

Weaknesses:

These have now been addressed in the revised version.

---

## [Author Response]

The following is the authors’ response to the original reviews

**Public Reviews:**

**Reviewer #1 (Public review):**
Summary:The authors use the teleost medaka as an animal model to study the effect of seasonal changes in day-length on feeding behaviour and oocyte production. They report a careful analysis of how day-length affects female medakas and a thorough molecular genetic analysis of genes potentially involved in this process. They show a detailed analysis of two genes and include a mutant analysis of one gene to support their conclusionsStrengths:The authors pick their animal model well and exploit the possibilities to examine in this laboratory model the effect of a key environmental influence, namely the seasonal changes of day-length. The phenotypic changes are carefully analysed and well-controlled. The mutational analysis of the agrp1 by a ko-mutant provides important evidence to support the conclusions. Thus this report exceeds previous findings on the function of agrp1 and npyb as regulators of food-intake and shows how in medaka these genes are involved in regulating the organismal response to an environmental change. It thus furthers our understanding of how animals react to key exogenous stimuli for adaptation.Weaknesses:The authors are too modest when it comes to underscoring the importance of their findings. Previous animal models used to study the effect of these neuropeptides on feeding behaviour have either lost or were most likely never sensitive to seasonal changes of day length. Considering the key importance of this parameter on many aspects of plant and animal life it could be better emphasised that a suitable animal model is at hand that permits this. The molecular characterization of the agrp1 ko-mutant that the authors have generated lacks some details that would help to appreciate the validity of the mutant phenotype. Additional data would help in this respect.

We would like to thank Reviewer #1 for the really constructive advice. In the revised manuscript, we provided more information on the molecular characterization of the *agrp1* KO-mutant and to emphasize the importance of our present animal model that permits the analysis of neuropeptide effects on feeding behavior in response to seasonal changes of day length.

**Reviewer #2 (Public review):**
Summary:The authors investigated the mechanisms behind breeding season-dependent feeding behavior using medaka, a well-known photoperiodic species, as a model. Through a combination of molecular, cellular, and behavioral analyses, including tests with mutants, they concluded that AgRP1 plays a central role in feeding behavior, mediated by ovarian estrogenic signals.Strengths:This study offers valuable insights into the neuroendocrine mechanisms that govern breeding season-dependent feeding behavior in medaka. The multidisciplinary approach, which includes molecular and physiological analyses, enhances the scientific contribution of the research.Weaknesses:While medaka is an appropriate model for studying seasonal breeding, the results presented are insufficient to fully support the authors' conclusions.Specifically, methods and data analyses are incomplete in justifying the primary claims:- the procedure for the food intake assay is unclear;- the sample size is very small;- the statistical analysis is not always adequate.Additionally, the discussion fails to consider the possible role of other hormones that may be involved in the feeding mechanism.

We would like to thank Reviewer #2 for the helpful comments. As the reviewer suggested, we revised the paragraph describing the procedure for the food intake assay to make it much easier for the readers to understand in the revised manuscript. In Figure 1-Supplementary figure 2, RNAseq was performed to search for the candidate neuropeptides, and that’s why the sample size was the minimum. On the other hand, each group in the other experiments consist of n ≥ 5 samples, which is usually accepted to be adequate sample size in various studies (cf. Kanda et al., Gen Comp Endocrinol., 2011, Spicer et al., Biol Reprod., 2017). As for the statistical analyses, we revised our manuscript so that the readers may be convinced with the validity of our statistical analyses.

**Reviewer #3 (Public review):**
Summary:Understanding the mechanisms whereby animals restrict the timing of their reproduction according to day length is a critical challenge given that many of the most relevant species for agriculture are strongly photoperiodic. However, the principal animal models capable of detailed genetic analysis do not respond to photoperiod so this has inevitably limited progress in this field. The fish model medaka occupies a uniquely powerful position since its reproduction is strictly restricted to long days and it also offers a wide range of genetic tools for exploring, in depth, various molecular and cellular control mechanisms.For these reasons, this manuscript by Tagui and colleagues is particularly valuable. It uses the medaka to explore links bridging photoperiod, feeding behaviour, and reproduction. The authors demonstrate that in female, but not male medaka, photoperiod-induced reproduction is associated with an increase in feeding, presumably explained by the high metabolic cost of producing eggs on a daily basis during the reproductive period. Using RNAseq analysis of the brain, they reveal that the expression of the neuropeptides agrp and npy that have been previously implicated in the regulation of feeding behaviour in mice are upregulated in the medaka brain during exposure to long photoperiod conditions. Unlike the situation in mice, these two neuropeptides are not co-expressed in medaka neurons, and food deprivation in medaka led to increases in agrp but also a decrease in npy expression. Furthermore, the situation in fish may be more complicated than in mice due to the presence of multiple gene paralogs for each neuropeptide. Exposure to long-day conditions increases agrp1 expression in medaka as the result of increases in the number of neurons expressing this neuropeptide, while the increase in npyb levels results from increased levels of expression in the same population of cells. Using ovariectomized medaka and in situ hybridization assays, the authors reveal that the regulation of agrp1 involves estrogen acting via the estrogen receptor esr2a. Finally, a loss of agrp1 function mutant is generated where the female mutants fail to show the characteristic increase in feeding associated with long-day enhanced reproduction as well as yielding reduced numbers of eggs during spawning.Strengths:This manuscript provides important foundational work for future investigations aiming to elucidate the coordination of photoperiod sensing, feeding activity, and reproduction function. The authors have used a combination of approaches with a genetic model that is particularly well suited to studying photoperiodic-dependent physiology and behaviour. The data are clear and the results are convincing and support the main conclusions drawn. The findings are relevant not only for understanding photopriodic responses but also provide more general insight into links between reproduction and feeding behaviour control.Weaknesses:Some experimental models used in this study, namely ovariectomized female fish and juvenile fish have not been analysed in terms of their feeding behaviour and so do not give a complete view of the position of this feeding regulatory mechanism in the context of reproduction status. Furthermore, the scope of the discussion section should be expanded to speculate on the functional significance of linking feeding behaviour control with reproductive function.

We would like to thank Reviewer #3 for the insightful advice. We added several pertinent sentences describing the ovariectomized female fish and juvenile fish, and our revised manuscript will give more complete view of their feeding regulatory mechanism in the context of reproduction status. In addition, we revised the discussion section to incorporate the valuable suggestion of the Reviewer #3.

**Recommendations for the authors:**

**Reviewer #1 (Recommendations for the authors):**
General: the text could profit from a careful editing of errors, including adjusting singular and plural status of nouns and verbs: examples are line 107 noun, line 96 verb suitable text editing software is available to do this task

Thank you for your suggestion. We thoroughly read the entire manuscript and corrected such errors in the revised manuscript.

As medaka is a unique genetic vertebrate model to study seasonal effects, it would be interesting to know whether the authors found novel or rather unexpected genes with a differential expression between LD and SD. It is understandable that the authors focused on argrp1 and npyb, as these have already been well studied in mammalian models although not in this context. Novel insights with genes previously not implicated in feeding regulation could underscore the unique nature of medaka as a model.

We appreciate your kind comments, which we found really encouraging to us. Since we focused on feeding-related peptides, we did not find any novel genes that have not been reported.

ISH is unreliable as a methodology to quantify expression levels. Yet the authors use this to compare fed and starved females to compare expression levels of agrp1. They use a temporal staining comparison and compare 90-minute and 300-minute staining reactions. However, they do not explain why they use the 90-minute staining time point and why 300 minutes of staining is the "saturation point of staining". They should provide compelling data for their claim and the selection of time points or else refrain from using these (at best) semi-quantitative ISH and provide more detailed (using serial sections) data to quantify the number of expressing cells.Anyhow, the quantification of mRNA expression levels may not be that significant when trying to compare different states of gene function, as translational and post-translational steps can have large effects on gene function. This should be discussed adequately.

Thank you very much for your comments. We conducted ISH by using medaka under LD or SD, not using those under fed or starved conditions. In addition, our previous study demonstrated that the slopes of the increase in the number of cells stained by ISH are also different if there is a difference in the expression level (Mitani et al., 2010). Although we do not have quantitative data of cell numbers, we confirmed that the number of cells expressing *agrp1* was saturated around 300 mins in our preliminary experiments, and therefore we terminated the chemogenic reactions at 300 mins. Based on these, we compared the cell ratio of 90 min (beginning of coloring) /300 min (saturation). However, since this analysis may not be worth discussing in detail, we moved this part to the supplementary figure as the reviewer suggested.

The molecular characterization of the agrp1 ko mutant is a bit thin.Line 221: "We obtained *agrp1*^−/−^ medaka, which has lots of amino acid changes in functional site for AgRP1" is a bit vague as a description for the ko-mutation. It would be really helpful if the authors could provide a scheme showing the wt protein with the relevant functional sites alongside the presumptive mutant protein.How did the authors verify the molecular nature of their mutation? They should use suitable antibodies and western-blot analysis (maybe reagents from Shainer et al., 2019 work in medaka); in case this is not possible they could isolate & clone the mutant transcript and use in-vitro translation systems to show that the presumptive mutant protein can actually be translated from this transcript. Another strategy could be to use a second non-allelic and (hopefully) non-complementing mutation (ko1/ko2 heterozygots for example) to show that ko-mutation acts the way the authors presume. The authors mention agrp1 ko medaka lines (plural!) in line 520, thus they may have an additional ko allele at hand.

Thank you very much for your comments. We explained the mutation site in Figure 6-Supplementary Figure 1 (A: DNA sequences and B: predicted amino acid sequence, of WT and mutants). In addition, we added immunohistochemistry data of WT and mutant using anti-AgRP antibody (Figure 6-Supplementary Figure 1C). While AgRP-immunoreactive signals were observed in WT, those were not in *agrp1*^−/−^. This result suggests that AgRP1 is not functional in *agrp1*^−/−^.

Presumably, the authors analysed heterozygous *agrp1*^+/−^ females and found they are as wt. If so the authors should say so.

Yes, we analyzed food intake of *agrp1*^+/−^. We added a supplementary figure (Figure 6-Supplementary Figure 2) and a sentence in L. 233-234.

How about *agrp1*^−/−^ medaka males: do they show a discernible phenotype?

We analyzed the phenotypes of *agrp1*^−/−^ males but did not describe the results, since the present paper only focused on female-specific feeding behavior.

*agrp1*^−/−^ females show no significant sensitivity of food intake to day length (Figure 6C). Does their (reduced) oocyte production react to day length? With other words: how much of the seasonal sensitivity is left in *agrp1*^−/−^ females. The authors suggest that E2 acts upstream of agrp1 and therefore some seasonality may still be left in *agrp1*^−/−^ females.

Although *agrp1*^−/−^ female is suggested to display abnormal seasonality of food intake, *agrp1*^−/−^ female in LD spawns and that in SD does not, indicating that seasonality of gonadal maturation still remains in *agrp1*^−/−^ female.

The authors show that *fshb* and *lhb* are downregulated in *agrp1*^−/−^ females. Is this also the case in wt females at SD?

Thank you very much for your comment. As described above, *agrp1*^−/−^ can spawn, which indicates that mechanisms for the downregulation of gonadotropins in *agrp1*^−/−^ may be different from that in SD female.

Figure 1_Supplementary Figure 2: the trends are visible in B and C, however, there is quite some variance between LD1, 2, and 3; the same for SD 1, 2, and 3. Can the authors give an explanation for this?

Since the data for LD1, 2, and 3 (SD1, 2, and 3) were obtained from different individual fish, the variance may be reasonable. We conducted expression analyses by using RNA-seq to find candidate genes that show larger differences than individual ones.

Figure 7E: the ovaries are difficult to see and the size bar in the wt picture is missing.

Thank you very much for your comments. We added a scale bar in the wt picture.

509 ff: the authors do not describe what exactly the "sham operation" encompasses: were the females just anesthetised or was there an actual operation without removing the ovaries?

The sham operation group was anesthetized, received an abdominal incision without removing the ovaries, and received skin suture by using a silk thread. We added this explanation in the Method section.

519 ff: was the *agrp1*^−/−^ ko induced in the d-rR strain to have the same genetic background as the wt fish?

Exactly. As the reviewer pointed out, the genetic background of *agrp1* -/- was the same as that of WT.

Minor points (Text edits):Line 42: change "when" into "where".Line: 54 "under the fixed appropriate ambient temperature" change into "while keeping an appropriate temperature constant".Line 55: here it would be good to briefly explain what long-day and short-day is so that the reader has an idea about the changes required without having to scroll down to the M&M section. For example LD 14/10 light-dark cycle, SD 10/14 light-dark cycle.Line 88: change "measurement" into "measuring".Line 96 change eats -> eat.Line 107 change female -> females.

We deeply appreciate the reviewer’s suggestions described above. We corrected them as the reviewer suggested (L. 42, L. 54, L. 55, L. 89, L. 96, L. 107).

Line 144-145: the sentence "since hypothalamic npy control..." does not make sense. Please correct.

Thank you very much for your suggestion. We corrected the sentence so that it makes sense (L. 145-146).

Line 180 and 185: the term here should be "LD induced sexual activity" rather than maturity. Age is the main determinant of maturity whereas light (LD) determines activity, in other words SD females are sexually mature if they are post-puberty stage.

Thank you very much for your suggestion. Since the sentence “LD-induced sexual maturity” made the reviewer confused, we corrected the sentence “substance(s) from LD-induced mature ovary” or “ovarian maturity”. Even though SD females are at post-puberty stage, their ovaries are immature and do not possess mature oocytes (L. 181).

Line 222: the authors should include the relevant information about the females: presumably agrp1.

In Line 226-228, we explained the phenotypes of *agrp1* knockout and added information for AgRP1 protein in Figure 6-Supplementary figure 1C.

Lines 449 ff: authors should state that the analysis was done in females, instead of just writing "medaka". This is also in line with the preceding paragraph of the M&M section.

Thank you very much for your suggestions. We corrected the sentence as the reviewer suggested (L.469)

Line 305: change like other mammals -> like in mammals.

Thank you very much for your suggestion. We corrected the sentence as the reviewer suggested (L. 320)

**Reviewer #2 (Recommendations for the authors):**
(1) The procedure of the food intake assay is not clear.- Habituation Period: Medaka were placed into a white cup containing 100 mL of water and allowed to habituate for 5 minutes. However, is 5 minutes sufficient to reduce stress in the fish? A stressed fish does not exhibit the same feeding behavior as an unstressed one.

Thank you for your comment. We confirmed that 5 minutes is enough for habituation in medaka, since medaka can swim freely in a few minutes after replacement from the tank and show normal feeding behavior.

- Feeding Protocol: Medaka were fed with 200 μL aliquots of brine shrimp-containing water. This procedure was repeated multiple times. How many times was this feeding procedure repeated? Was it 3, 10, or 100 times?

Although there was a small variation in each trial, we usually applied tubes about 5 times or so.

- Brine Shrimp Counting: You collected 10 mL of the breeding water to count the number of uneaten brine shrimp. Can you confirm that sampling 10% of the total volume is representative? Were any tests conducted to validate this? Given that you developed an automated tool to count the brine shrimp, why didn't you count them in all 100 mL?

The reason for collecting 10 mL is to collect the leftover shrimp as soon as possible. Ten mins after the start of the experiment, we quickly placed a magnetic bar to stir the breeding water so that the shrimp concentration will be constant. Then we collected 10 mL aliquot from the experimental cup by using a micro pipette. In preliminary trials, we applied shrimps, the amount of which is almost the same as that applied to WT medaka in LD, to a white cup containing 100 mL water, and we divided it into 10 mL and 90 mL aliquots and separately counted the number of shrimps in each aliquot. Here, we confirmed that the variance between the numbers calculated by counting the shrimps in 10 mL aliquot and the total volume of 100 mL falls within the range of the variance of total applied shrimp. Thus, our present counting method can be considered reasonable.

- Brine Shrimp Aliquot Measurement: You mentioned counting the number of brine shrimp in the 200 μL solution three times before and after the experiments. What does this mean? Did you use this procedure to calculate the mean number of brine shrimp in each 200 μL aliquot?

Thank you for your comment. As the reviewer commented, to calculate the mean number of brine shrimp in each 200 µL aliquot, we counted the number of brine shrimp in the 200 µL solution three times before and after the experiments.

- How did you normalize the food intake data? This procedure is not detailed in the methods section.

Thank you very much for pointing it out. We normalized food intake by subtracting the amount of shrimp by the average of those in LD or WT fish. This explanation was added in the Method section (L. 439).

(2) Sample Size. Various tests were conducted with a low number of medaka (e.g., 2 brains for RNA-seq, 8 females for ovariectomy). Are these sample sizes sufficient to draw reliable conclusions?

In Figure 1-Supplementary figure 2, RNAseq was performed to search for the candidate neuropeptides, and that’s why the sample size was the minimum; we pooled two brains as one sample and used three samples per group. On the other hand, each group in the other experiments consist of n ≥ 5 samples, which is usually accepted to be adequate sample size in various studies (cf. Kanda et al., Gen Comp Endocrinol., 2011, Spicer et al., Biol Reprod., 2017).

(3) Statistical Analysis.- The authors used both parametric and non-parametric tests but did not specify how they assessed the normal distribution of the data. For example, if I understood correctly, a t-test was used to compare a small dataset (n=3). In such cases, a U-test would be more appropriate.

Thank you for your comment. As for Figure 1 -Supplementary Figure 2C, we showed the graphs just to show you candidates. To avoid misunderstanding, we deleted statistical statements in that panel.

- It is unclear why the Steel-Dwass test was used instead of the Kruskal-Wallis test for comparing agrp1 and npyb expressions in control, OVX, and E2-administered medaka.

While the authors mentioned using non-parametric tests, they did not specify in which contexts or conditions they were applied.

Thank you very much for your comment. Kruskal-Wallis test statistically shows whether or not there are differences among any of three groups. To perform multiple comparisons among the three groups, we used Steel-Dwass test.

- The results section lacks details on the statistical tests used, including the specific test (e.g., Z, U, or W values) and degrees of freedom.

Thank you for your comment. As the reviewer pointed out, we added such statements in all the figure legends containing statistics.

(4) Previous studies have shown that photoperiod treatments alter the production of various hormones in medaka (e.g., Lucon-Xiccato et al., 2022; Shimmura et al., 2017), some of which, like growth hormone (GH), have been shown to influence feeding behavior (Canosa et al., 2007).In your RNA-seq analysis, did you observe any changes in the expression of genes involved in other hormone synthesis pathways, such as pituitary hormones (GH and TSH), leptin, or ghrelin (e.g., see Volkoff, 2016; Blanco, 2020; Bertolucci et al., 2019)?Including such evidence in the discussion would provide a broader perspective on the hormonal regulation of food intake in medaka.

We appreciate your constructive comments. Unfortunately, since we performed RNA-seq using the whole brain after removal of the pituitary, we could not check such changes in the expression of pituitary hormone-related genes. As additional information about the feeding-related hormones, leptin did not show significant difference in our RNA-seq analysis, and we could not analyze ghrelin because ghrelin has not been annotated in medaka (NCBI and ensembl).

**Reviewer #3 (Recommendations for the authors):**
There are some parts of the study that need to be developed further in order to provide a more comprehensive analysis.(1) In the juvenile as well as ovariectomized female fish, the authors should confirm experimentally whether day length influences feeding activity.

Thank you very much for your suggestion. We analyzed feeding behavior of juvenile (Figure 4-Supplementary Figure 1) and OVX female (Figure 5-Supplementary Figure 1). As shown in these figures, food intake in juvenile and OVX were not significantly different between LD and SD.

(2) More discussion as to the relevance of increasing feeding activity to support reproductive functions such as sustained egg production would be valuable. One assumes the metabolic costs of producing eggs on a daily basis in this species would inevitably require increased food intake. Is this a reasonable prediction?

We deeply appreciate your suggestion. We strongly agree with this argument, and we added such discussion in “Discussion” section (L. 406-408).

**Editor's note:**
Should you choose to revise your manuscript, if you have not already done so, please include full statistical reporting including exact p-values wherever possible alongside the summary statistics (test statistic and df) and 95% confidence intervals. These should be reported for all key questions and not only when the p-value is less than 0.05 in the main manuscript.

We appreciate the editor’s suggestion. We added P-value in the main manuscript, where statistical analyses were performed. In addition, we described test statics in the figure legends. We did not use df values for the statistics used in the present analyses, and therefore did not describe it in the main text.